# Gas-Dynamic Kinetics of Vapour Sampling in the Detection of Explosives

**DOI:** 10.3390/molecules24234409

**Published:** 2019-12-03

**Authors:** Vladimir M. Gruznov, Alexander B. Vorozhtsov

**Affiliations:** 1Trofimuk Institute of Petroleum Geology and Geophysics of Siberian Branch of Russian Academy of Sciences, Koptug Avenue, 3, Novosibirsk 630090, Russia; 2Analytical Chemistry Department, Novosibirsk State University, Pirogova St., 2, Novosibirsk 630090, Russia; 3High-Energy and Special Materials Research Laboratory, National Research Tomsk State University, Lenin Avenue, 36, Tomsk 634050, Russia; abv1953@mail.ru

**Keywords:** dynamics of sorption concentration, rapid vapour concentration, complete vapour capture, vapour transportation

## Abstract

The dynamic sorption concentration of explosive vapours on concentrators made of a metal mesh, and the transportation of explosive vapours through the extended metal channels are considered. The efficiency of the concentration and transportation is determined by the breakthrough of the substance’s molecules through the channels. The research methods we used were breakthrough calculation theory and experiment. When calculating the breakthrough, a mesh was presented as a set of parallel identical channels. Wire mesh and extended channels were made of stainless steel. The breakthrough is determined through the specific frequency of the collisions between the molecules and the channel’s surface. This is presented as a function of the ratio of the substance diffusion flow to the channel’s surface to the airflow through the channel. The conditions for high-speed concentration, complete capture of explosive vapours, and low vapour losses during their transportation through the extended channels were determined theoretically and experimentally. For a concentrator made of a mesh, the condition of a high concentration rate at a high breakthrough (up to 80%) was determined. The described sorption concentration is used in portable gas chromatographic detectors of explosive vapours of the EKHO series.

## 1. Introduction

In the anti-terrorist gas-analytical control of objects for the content of explosives by smell using a gas analytical sampling method, there is the problem of effective rapid sampling of the air with explosive vapours near the objects. We note that there is no such problem in remote methods [1,2]. In sampling methods several modes of vapour sampling are possible depending on the problem being solved and detection conditions, namely: (i) rapid accumulation of a substance at the concentrator in the detection of explosive vapours; (ii) complete vapour capture at the concentrator when determining the vapour concentration; (iii) mode of small vapour losses during the vapour sampling through the extended channels.

For rapid accumulation of a substance at the concentrator in [3,4], incomplete vapour capture at a concentrator made of a metal mesh was used. The incomplete vapour capture is characterized by the value of the breakthrough β of molecules through a concentrator. The breakthrough is determined by the ratio of the number *N* of molecules that were not captured by the concentrator to the number *N*_0_ of molecules that entered the channel. It was shown that a high rate of explosive vapour accumulation is achieved at a high breakthrough of up to 80%.

The complete capture mode is used to quantify the concentration of explosive vapours in the air near the controlled object. For this mode, the breakthrough value is close to zero. In [5], the condition for the complete capture of TNT vapours by a concentrator made of a metal mesh when defining the concentration of TNT vapour was experimentally determined: a sample of 1 L of TNT vapour at the initial concentration of 8.1 × 10^−14^ g/cm^3^ to the concentrator within 1 min. Hereafter, TNT vapour concentrations are given at room temperature +25 °C. If it is necessary to take into account the temperature dependence of the concentration of TNT vapours, one can use the results from [6].

The mode of small vapour losses during the vapour sampling through the extended channels was also used in [5] for the automated control of hand luggage in an automatic storage unit. For this mode, the breakthrough value is close to 1.

In all three cases of vapour sampling, the value of the breakthrough of explosive molecules through a concentrator or extended channels plays a crucial role in achieving the goal.

The purpose of the article is based on the previously developed calculation model of the breakthrough of explosive vapours through a concentrator in the form of an extended channel, to show the commonality of the gas-dynamic approach to the characterization of the processes: rapid accumulation, complete capture, and transportation of explosive vapour through a sampling channel. The kinetics of sampling is understood as the temporal patterns of the processes of explosive vapours’ capture and transportation. Trinitrotoluene was used as an explosive.

When calculating the breakthrough, a mesh was presented as a set of parallel identical round channels. Wire mesh and extended channels were made of stainless steel.

## 2. Results

### 2.1. Theoretical Section

The breakthrough calculation for a concentrator consisting of one channel. First, we consider a model of a concentrator in the form of a single metal channel (tube) with an internal radius r, length l. The scheme of the concentrator’s longitudinal section is shown in Figure 1. A flow q of gas (air) containing molecules of the substance being concentrated is passed through the channel. We consider a laminar gas flow mode. With the gas flow, the number *N*_0_ of molecules of the substance enters the channel’s inlet, some of the molecules are adsorbed, and *N* molecules exit the channel. We determine the value of the breakthrough β equal to:β = *N/N*_0_(1)

The breakthrough is determined using the concepts of vibrational relaxation of adsorbed molecules on the channel’s surface [7]. According to this theory, the adsorption rate *dθ*/*dt* of molecules per unit area of the channel’s surface can be determined as proportional to the specific frequency *Z*, cm^−2^·s^−1^, of the collisions between the molecules and the surface of the channel:*dθ/dt = sZ*(2)
where *θ* is the number of molecules adsorbed per unit area of the channel’s surface, *s* is the coefficient (probability) of “adhesion” of molecules to the sorbent surface, determined by the thermodynamics of the interaction of molecules with the channel’s surface material.

The number *N*t of molecules adsorbed by the channel’s surface with area *F* is *N*t = *θF*. Then *N*_0_ = *N*t *+ N*. If *N*_0_ = *const*, we obtain: *dN*t = −*dN*. Thus, we can write that:*dN/dt = −sZF*(3)

Under the condition that *dN*t = −*dN*, we define the frequency *ZF* of collisions of molecules with the surface is proportional to the number *N* of molecules that passed through the channel and inversely proportional to the time $\tau_{d}$ of diffusion of molecules from the centre of the channel to the channel’s wall. Then the right part of the Equation (3) can be represented as:(4)$$sZF=sN/\tau_{d}$$
where $\tau_{d}$ is the channel time constant. Hence, *dN*/*d*t *= −sN*/$\tau_{d}$, and therefore
(5)$$N/N_{0}=\beta =exp(-st/\tau_{d})$$
where *t* is the time determined by the linear airflow rate in the channel. With a flow rate *q*, a length *l*, and a channel cross-section *S*, we have *t = lS*/*q*.

The time constant $\tau_{d}$ is determined through the diffusion coefficient *D* from the Einstein equation (Δ^2^ = 2*n*_d_
*D*$\tau_{d}$):(6)$$\tau_{d}=(1/2)(\Delta {)}^{2}/D/n_{d}$$
where *n*_d_ is the number of degrees of freedom, Δ is the root mean square displacement of the diffusion front. For Δ = r and *n*_d_ = 3, the function β that determines the breakthrough takes the form
β = *exp*(−*q*_d_/*q*)(7)
where q_d_ = 6π*Dls*. The physical sense of q_d_ is the diffusion flow of the substance being concentrated to the surface of the channel.

The breakthrough calculation for a concentrator consisting of n round parallel channels. In this case, we use the Equation (7) for the breakthrough through one channel. For a concentrator consisting of n identical channels, the breakthrough calculation formula takes the form
β = *exp*(*−Q*_d_*/Q*)(8)
where *Q* = *nq* is the airflow through *n* channels of the concentrator, *Q*_d_ = 6π*Dnls* is the diffusion flow of the substance being concentrated to the surface of all *n* channels.

### 2.2. Experimental Section

#### 2.2.1. Concentrators

We used concentrators made of a wire mesh. Wire with a diameter *d*_p_ = 0.05 mm was made of stainless steel. The mesh consists of *n* square cells (channels). The side of the square cell *b* was 0.08 mm. The scheme of the concentrator and the square cells’ dimensions are shown in Figure 2. The mesh diameter was 7.5 mm and the number *n* of square channels was about 2600.

To calculate the breakthrough through the mesh, the set of square cells (channels) of the mesh was replaced by a set of *n* round parallel channels with a radius *r* = *b*/2 and equivalent length *l*_c_. The equivalent length *l*_c_ of the channel in the mesh was determined from the condition that the calculated and experimental values of the gas-dynamic resistance *R* of the concentrator are equal. The value of *R* was calculated from the Poiseuille equation: *R* = 8*η**l_c_*/(π*nr*^4^); experimentally, it was determined by the equation *R*_0_ = *P*/*Q*, where *η* is the dynamic viscosity of the air, and *P* and *Q* are the experimental values of the pressure difference and airflow in the air pumping line in the presence of a concentrator, respectively, as shown in Figure 3. For the mesh that we used, the condition of *R* = *R*_0_ with an error not more than 10%, as described in [3], was fulfilled at *l_c_* = 2π*d_p_*. This value of length *l*_c_ was used in the value calculations of the breakthrough of molecules through a concentrator made of a mesh.

#### 2.2.2. Experimental Determination of the Breakthrough

To determine the breakthrough, two identical concentrators were installed in the air pumping line (Figure 3). The breakthrough was determined by the ratio of signals from the second (downstream) concentrator to the signal from the first concentrator. Since it is impossible to choose absolutely identical concentrators, the concentrators were interchanged and the breakthrough measurement was repeated. Thus, an average response was obtained for both concentrators.

To measure signals from concentrators, we used portable express multicapillary gas chromatographs (GC) of the EKHO series containing a device for sample injection from the considered mesh concentrators [3,5,8]. Express and highly sensitive chromatographs are designed to detect explosives. High sensitivity is provided by the rapid concentration of the substances on the metal mesh. Thermal desorption sample injection from mesh concentrators is carried out within 0.5 s, rapid separation is carried out within 15–30 s on a multicapillary column.

To determine the breakthrough, we used a multicapillary gas chromatographic detector of explosives EKHO-M [3] with the electron capture detector. The carrier gas was argon. EKHO-M threshold sensitivity for TNT vapour is 8–10 pg in the sample. An example of a chromatogram of TNT isomers mixture vapours captured by a concentrator is shown in Figure 4. The mass of 2,4,6-TNT isomer recorded from a concentrator is about 120 pg.

The experimental confirmation of the complete capture mode and vapours transportation through the extended channels was carried out with the analysis of samples on the EKHO-IMIS multicapillary gas chromatograph with an ion mobility increment spectrometer (IMIS) as a detector [5]. The carrier gas was atmospheric air cleaned by an integrated filter. This chromatograph is characterized by a high threshold sensitivity for TNT vapours (less than 8 pg in the sample). High sensitivity was demonstrated in [5] due to the application of the mode of complete capture of molecules on a mesh concentrator described in the article and a selective IMIS detector. High threshold sensitivity is necessary for effective TNT detection in luggage in an automatic storage unit.

## 3. Discussion

### 3.1. Breakthrough Definition

Figure 5 shows the calculated dependence according to the Equation (8) for two values of *Q*_d_—130 cm^3^/s (graph 1) and 140 cm^3^/s (graph 2), and the results of the breakthrough measurement using the EKHO-M device [3] under normal climatic conditions. The concentrator contained two meshes. In the calculations *D* = 0.2 cm^2^/s, *n* = 2600, *l* = 0.028 cm. For these data, at *s* = 0.5, the diffusion flow *Q*_d_ = 140 cm^3^/s, and *Q*_d_ = 130 cm^3^/s corresponds to *s* = 0.46.

In general, the simple breakthrough calculation Equation (8), obtained by the phenomenological method, satisfactorily describes the complex process of substances’ capture by a mesh concentrator.

### 3.2. Express Accumulation of the Substance on the Mesh Concentrator

As already noted, this mode is necessary for the rapid detection of explosive vapours near the surface of controlled objects. In this case, the problem of quickly sampling vapours in order to detect them without having to determine the concentration is solved.

Following [9], the mass *m* of the substance captured by the concentrator, taking into account the breakthrough, can be determined as
*m* = *Ct*_n_*Q*(1 − β)(9)
where *C* is the initial vapour concentration, *t*_n_ is the accumulation time, *Q* is the flow through the concentrator. Noting Equation (8) for β, we obtain
*m* = *Ct*_n_*Q*[1 − *exp*(−*Q*_d_/*Q*)](10)

In a dimensionless form, the efficiency of vapour mass accumulation depending on the flow is equal to
*m*/*m*_d_ = (*Q*/*Q*_d_)[1 − *exp*(−*Q*_d_/*Q*)](11)
where *m*_d_ = *CQ*_d_*t*_n_—is the maximum amount of substance captured by the concentrator with the given *Q*_d_, *t*_n_.

Figure 6 shows the dependence of *m/m*_d_ on the *Q*/*Q*_d_ ratio. The graph illustrates an almost linear increase in mass with increasing flow *Q* to about *Q*_d_. It can be seen here that too small and too large *Q*/*Q*_d_ values are disadvantageous, because at *Q*/*Q*_d_ < 1, the amount of captured substance sharply decreases, and at *Q*/*Q*_d_ ˃ 4, an increase in flow rate does not give a noticeable increase in the captured mass. The increase becomes less than 2% of its maximum value at *Q* → 0.

From Figure 6, it follows that the rational relative flow *Q/Q*_d_ lies in the range from two to four. Within this interval, there is a high accumulation efficiency in the range of 80–90% and a high (up to 80%) breakthrough β = 0.6÷0.8. The correspondence of the rational area of the *Q*/*Q*_d_ ratio to the area of effective accumulation of the substance is shown in grey in Figure 6.

The high mass accumulation rate at a high breakthrough is physically explained by the fact that mainly molecules located near the surface reach the surface of the concentrator due to diffusion, and a large flow ensures a high speed of delivery of surface molecules to the concentrator.

The express accumulation technology described above is implemented in EKHO type detectors [3,5,8]. For the two-mesh concentrator with its parameters: *n* = 2600, *l* = 0.028 cm, *s* = 0.5 and *D* = 0.2 cm^2^/s, flow *Q* = 280 cm^3^/s, calculation according to Equation (11) shows that, for the initial vapour concentration of 10^−14^ g/cm^3^, a sufficient for detection amount of a substance is accumulated in 10 seconds (about 11 pg). This occurs at the ratio *Q/Q*_d_ = 2 and the breakthrough of about 0.6.

### 3.3. Complete Vapour Capture Mode at the Concentrator

This mode is realized when the molecules during movement with the flow *Q* in the channel manage to reach its surface due to diffusion. From the Einstein Equation (6) we can determine the time of the mean square movement of the molecules by the value of the channel diameter so that all molecules reach its surface. From the condition that this time is shorter than the time of passage of the channel length by the molecules, we can estimate the flow condition for complete vapour capture. Indeed, for one channel, the full capture condition is as follows
4/(6*D*) < π*l*s/*q*(12)

For *n* channels
*Q* < ¼6π*Dnl*s = *Q*_d_/4; i.e., *Q* < *Q*_d_/4(13)

This means that, for example, for a two-mesh concentrator, complete capture will occur at *Q* < 2.1 L/min and with a breakthrough β < 0.02. Observing this condition at a flow of 1 L/min as seen in [5], almost complete capture was confirmed experimentally during concentration of TNT vapours in the air at a concentration of 8 × 10^−14^ g/cm^3^ and sampling to a concentrator of up to 20 pg of TNT, i.e., when passing of up to 250 mL of air sample through a mesh concentrator.

### 3.4. Vapours Transportation Through the Channels with Small Losses

To automate the monitoring of hand luggage in automatic storage units for the presence of explosives, sampling through a system of extended channels is used [5]. In this case, to reduce the vapour losses in the channel, a high vapour breakthrough via the channel is required, up to 1. From Equation (7) we can determine: ln β = −(*D*/*Q*)6π*l*s, where ln is the natural logarithm. We set *s* = 1. Then, for a given length *l* and channel radius *r*, a high breakthrough is achieved by selecting the *D*/*Q* ratio. To reduce losses, the following condition must be met:(*D*/*q*) < −ln β/(6π*l*)(14)

For example, with β = 0.9, *l* = 2.5 m, the following condition must be fulfilled for the *D*/*Q* ratio: (*D*/*q*) < 2 × 10^−5^, from which, in particular, it follows that for a TNT molecule diffusion coefficient *D* = 0.2 cm^2^/s, it is necessary to choose a *q* flow of about 10 L/s. The use of a flow of 6 L/s in [5] showed satisfactory results for the transportation of vapours through a duct system. From Equation (14), the condition on the channel length is also determined under other specified conditions:*l* < −ln β*/(D*/*q)*/(*6*π)(15)

## 4. Conclusions

The described phenomenological (or microscopic) calculation of the breakthrough of molecules through the channel defines the breakthrough as the ratio of the diffusion flow *Q*_d_ of the substance being concentrated to the surface of the concentrator to the flow *Q* of air with the vapour of the substance through the concentrator. The essence of this phenomenological approach to calculating the breakthrough is that the adsorption rate of molecules per unit area of the concentrator is determined as proportional to the specific frequency of collisions of the molecules with the surface. The proportionality coefficient or the coefficient of “adhesion” of molecules to the surface is also introduced. As a result, the breakthrough is defined as a function of the ratio of the diffusion flow of the substance to the channel’s surface to the airflow through the concentrator.

Depending on the breakthrough value, the following concentration modes are determined:express accumulation of a substance at the concentrator (breakthrough is up to 80%);complete vapour capture at the concentrator (breakthrough is close to zero);small vapour losses (breakthrough is close to 1) during the vapour sampling through the extended channels.

Thus, the use of the breakthrough demonstrates the commonality of the gas-dynamic approach to the characterization of various concentration modes.

It is also shown that the gas-dynamic characteristic of the modes is also determined by the ratio of the diffusion coefficient *D* of the substance’s molecules to the flow *Q* (for one channel *Q* = *q*). The *D/Q* ratio plays the role of a certain invariant that determines the concentration mode. The Table 1 below illustrates the high “sensitivity” of the *D/Q* ratio to the mode type.

The shown dependences of the concentration modes on the breakthrough and the *D/Q* ratio can help developers to optimize the geometry of the concentrators (by choosing the length of the channels, their number, section, etc.,), and the temperature operation modes of the concentrators (*D*, specific frequency of the collisions *Z* and *s* depend on the temperature).

The described approach to determining the parameters of mesh concentrators for the modes of express accumulation of substance on a concentrator and complete capture was used in portable gas chromatographic detectors of explosive vapours of the EKHO series [3,5,8].

In conclusion, we note that the extremely simple, neat calculation of Equation (8) for the breakthrough satisfactorily describes the complex process of substance capture by a concentrator. It can also be used to evaluate the adhesion coefficient (probability) *s* and to measure the diffusion coefficient.

## Figures and Tables

**Figure 1 molecules-24-04409-f001:**
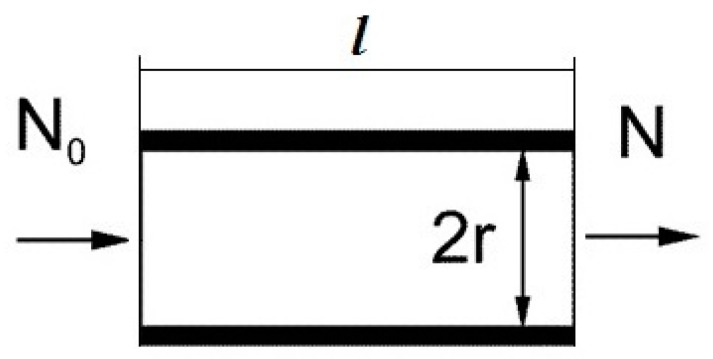
Scheme of sampling channel.

**Figure 2 molecules-24-04409-f002:**
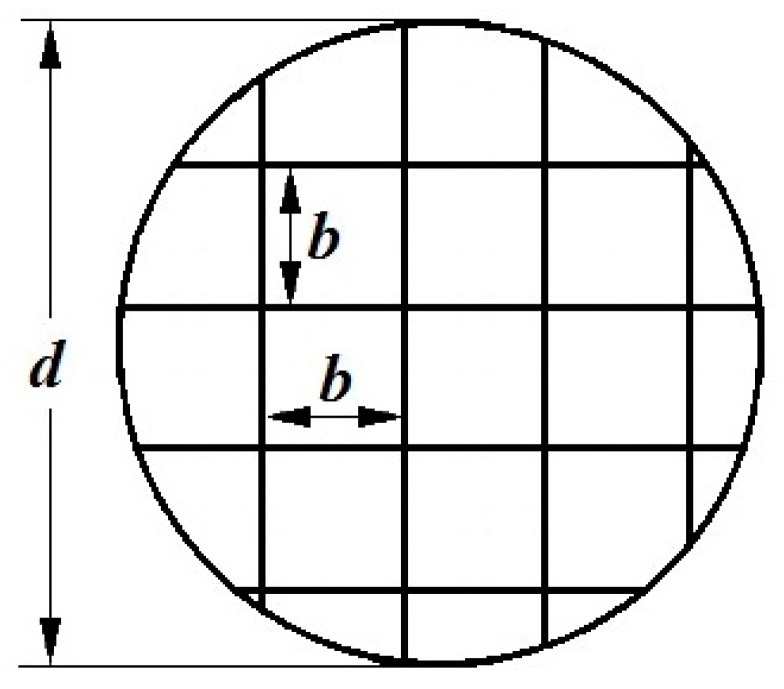
Scheme of a metal mesh concentrator, where *d*—concentrator mesh diameter, *b*—side of a square cell (channel) of the mesh.

**Figure 3 molecules-24-04409-f003:**
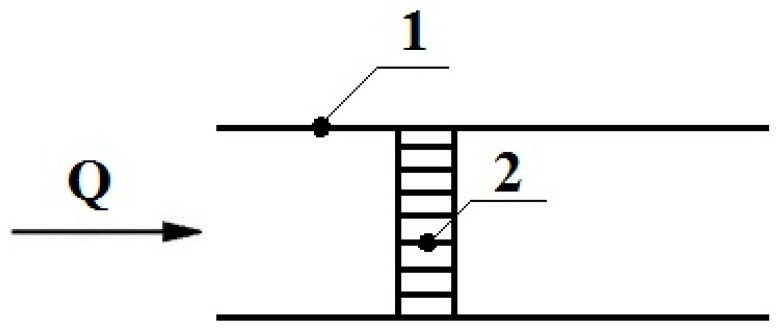
Scheme for determination of the gas-dynamic resistance of a concentrator, where 1 is a line for pumping air through a concentrator, 2—a concentrator.

**Figure 4 molecules-24-04409-f004:**
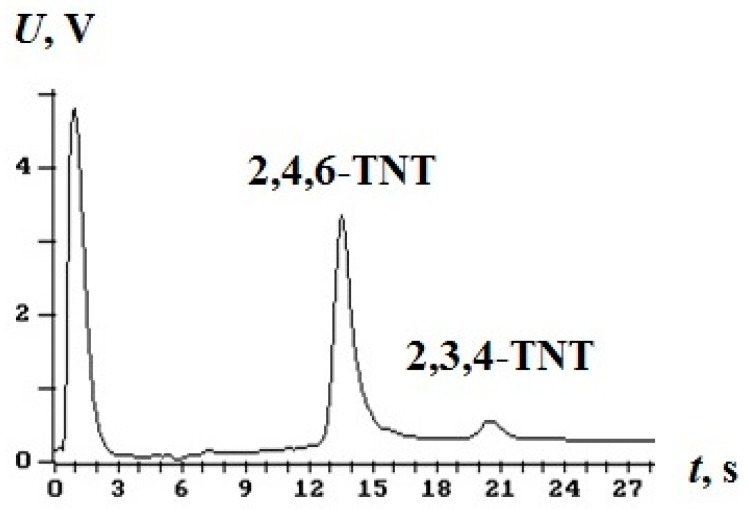
Chromatogram of vapours of TNT isomers.

**Figure 5 molecules-24-04409-f005:**
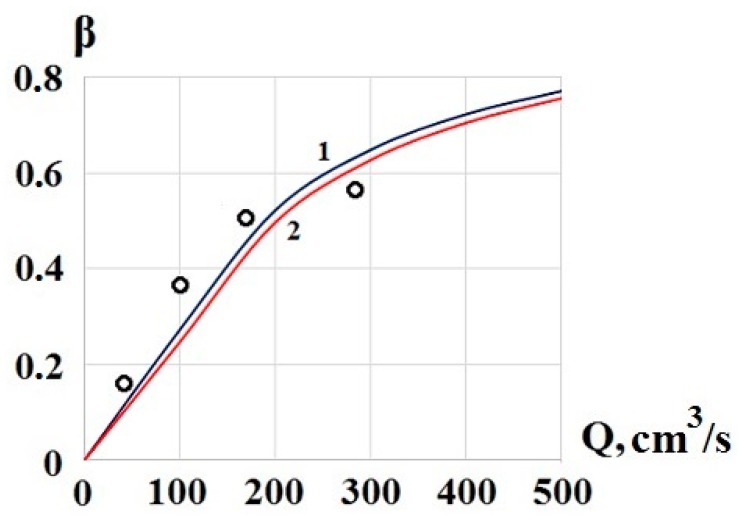
Dependence of the breakthrough of TNT molecules through the concentrator on airflow. Designations: calculation 1 for *Q*_d_ = 130 cm^3^/s, 2 for *Q*_d_ = 140 cm^3^/s, O, measurements.

**Figure 6 molecules-24-04409-f006:**
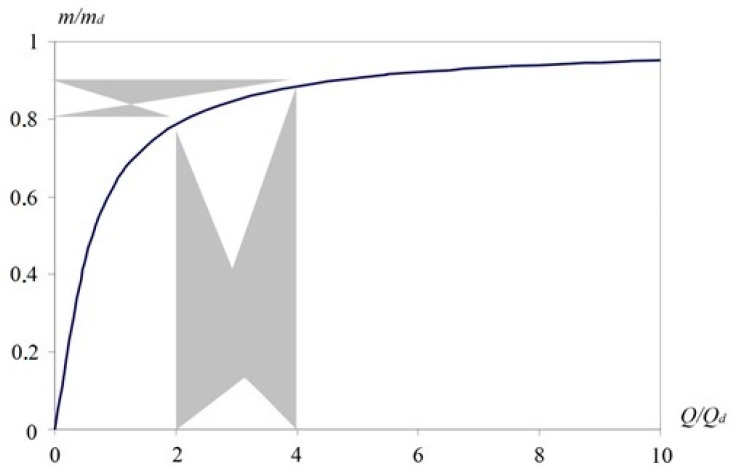
The effectiveness of the substance accumulation at the concentrator depending on airflow.

**Table 1 molecules-24-04409-t001:** Concentration modes.

Mode Type	Breakthrough	*D*/*Q* Ratio, cm^−1^
Vapour transportation	0.9	2 × 10^−5^
Rapid substance accumulation	0.6	7 × 10^−4^
Full capture	0.02	6 × 10^−3^

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
