# Peer review of "Gas-Dynamic Kinetics of Vapour Sampling in the Detection of Explosives"

_molecules, 2019, doi:10.3390/molecules24234409_

Round 1

Reviewer 1 Report

Dear Authors,

After a careful reading of the article, I found the article very interesting.

The conditions for high-speed concentration, complete capture of explosives vapors and low vapor losses during their transportation through channels were determined theoretically and experimentally. All the theoretical assumptions were clearly validated and explain using an appropriate mathematic approach. This study is a useful asset to help other researchers to understand and to optimize the geometry of the concentrator, for example to choose an appropriate length of the channel, their number and section.

The introduction part needs to be more focused and comprehensive in all the previous studies focused, it is not clear and confusion for the reader.

However, I don´t think this article is in line with the scopes of the Journal Molecules. I consider it more appropriate for this paper to submit in a Journal more focus on Sensors or/and design sensors.

Author Response

Dear Reviewer!

We sincerely thank You for Your comments for improving our paper!

The text of the paper was adjusted in accordance with the comments.

In the introduction, we have clarified the status of the problem and the purpose of the article.

Also, in the theoretical part, we have made some additional explanations and we have added some more illustrations to the experimental part. 

Sincerely Yours,

 Vladimir M. Gruznov,

 Alexander B. Vorozhtsov

Reviewer 2 Report

This paper considers the optimisation of vapour sampling in a preconcentrator for trace explosives detection. A simple model is derived to find expressions that describe the design conditions for three scenarios: (1) where rapid accumulation of a substance is required at the concentrator; (2) where complete vapour capture atthe concentrator is required; (3) where small vapour losses during sampling through extended channels are required.

Overall I think this is an interesting paper worthy of publication in Molecules, which would be of interest to researchers working in this field. Prior to acceptance, however I have a number of minor corrections which the authors should address.

Section 1- Introduction: The introduction to the paper could be more clearly written to explain the context and what the paper plans to achieve. Currently the introduction text is rather impenetrable.

Line 94- why is the equation labelled (3*)? I suggest this be renumbered to (4) and all subsequent equations are renumbered accordingly.

Line 109 – “the square cell (channel) at a clearance of 0.08 mm, were considered": please explain more clearly what this sentence means. It would be very helpful to add a diagram which shows the geometry of the mesh which is considered in this paper.

Line 128 – “the equivalent channel length l in a single mesh was taken equal to… in accordance with [3].” Please explain the sentence more clearly so that this section can be understood without a requirement to refer to reference [3]. Again this would benefit from a diagram which shows the geometry of the mesh which is considered in this paper.

Author Response

Dear Reviewer!

We sincerely thank You for Your comments for improving our paper!

The text of the paper was adjusted in accordance with the comments.

1.In the introduction, we have clarified the status of the problem and the purpose of the article. 

2.In the theoretical part, we have made some additional explanations. We have introduced the sequential numbering of formulas and excluded the designation (3*).

3.We have added some illustrations to the experimental part.

Figure 2 illustrates the scheme of a concentrator. Figure 3 illustrates the method for determining the equivalent channel length and experimental method for the breakthrough definition. Figure 4 is a chromatogram that illustrates the high quality of registration of the vapour captured by a concentrator and a high-speed response of the gas chromatograph.

Also, we have made some explanations to the existing figures: the curves in Figure 5 are made in different colours; in Figure 6, the designation of grey areas is explained.

Sincerely Yours,

 Vladimir M. Gruznov,

 Alexander B. Vorozhtsov

Reviewer 3 Report

The manuscript describes the kinetics of explosives vapours with respect to a concentrator prior to analysis by gas chromatography. The kinetics are modelled mathematically prior to being described experimentally. The research appears to contribute interesting results to the field, though there are aspects of the paper I recommend improving prior to publication.

1) "Vapor" in the title is "vapour" elsewhere in the manuscript.

2) Line 12 - is "long" referring to the 0.08mm channel? I'd describe that as short.

3) Figure 1 seems to be missing some descriptors - e.g. are the two circles inside the rectangle molecules?

4) The experimental section needs substantially more detail for this to be reproducible. Some extra diagrams or labelled photos would be useful. There is very little detail on the experimental set-up beyond references to the authors' previous work - the paper therefore struggles to stand alone. 

5) Figure 2 could be presented a bit clearer - i.e. the two curves could be different colours and with a legend stating Qd = 130 cm3/s etc.

6) It's not clear in Figure 3 what the grey shaded areas are.

7) The stated record of 0.08 pg is not very convincing due to the lack of experimental detail.

8) More references should be included beyond self-citations.

Author Response

Dear Reviewer!

We sincerely thank You for Your comments for improving our paper!

The text of the paper was adjusted in accordance with the comments.

We have tried our best to improve the language of the paper.

We have added some more figures to the experimental part. Figure 2 illustrates the scheme of a concentrator. Figure 3 illustrates the method for determining the equivalent channel length and experimental method for the breakthrough definition. Figure 4 is the chromatogram that illustrates the high quality of registration of the vapour captured by a concentrator and a high-speed response of the gas chromatograph.

The curves in Figure 5 (Figure 2 in the previous version) are made in different colours.

In Figure 6 (Figure 3 in the previous version) the designation of grey areas is explained.

Also, in the introduction, we have clarified the status of the problem and the purpose of the article. In the theoretical part, we have made some additional explanations. We have introduced the sequential numbering of formulas and excluded the designation (3*).

A detailed description of gas chromatographs was not given, because this is not the purpose of the article. If there is an interest in the method of achieving a record sensitivity of 0.08 pg in a sample, we can prepare a separate article on multicapillary gas chromatography.

Sincerely Yours,

 Vladimir M. Gruznov,

 Alexander B. Vorozhtsov

Round 2

Reviewer 3 Report

The paper is now much clearer.